# Endogenous Hypersensitivity Infection: A Unifying Framework for *Cutibacterium acnes*-Associated Sarcoidosis

**DOI:** 10.3390/microorganisms14010147

**Published:** 2026-01-09

**Authors:** Yoshinobu Eishi

**Affiliations:** Department of Human Pathology, Graduate School of Medical and Dental Sciences, Institute of Science Tokyo, 1-5-45 Yushima, Bunkyo-ku, Tokyo 113-8519, Japan; eishi.path@tmd.ac.jp or eishi.yoshi@gmail.com

**Keywords:** sarcoidosis, *Cutibacterium acnes* (formerly *Propionibacterium acnes*), endogenous hypersensitivity infection (EHI), latent intracellular infection, immune tolerance failure, regulatory T cells, Th1/Th17 inflammation, host–microbe interaction

## Abstract

Sarcoidosis is an immune-mediated granulomatous disease whose etiology has remained unresolved despite more than a century of investigation. Accumulating microbiological and immunopathological evidence now implicates *Cutibacterium acnes*—a ubiquitous indigenous commensal—as the most consistent antigenic trigger. Its frequent detection within sarcoid granulomas by quantitative PCR, in situ hybridization, and species-specific immunohistochemistry suggests latent intracellular persistence and the potential for endogenous reactivation. To explain how a noncontagious commensal can drive granulomatous inflammation, this review proposes the concept of Endogenous Hypersensitivity Infection (EHI). EHI describes a host-centered process in which reactivation of latent intracellular microbes leads to the breakdown of immune tolerance and provokes Th1-dominant hypersensitivity responses in genetically and immunologically susceptible individuals. This framework bridges the traditional divide between infection and autoimmunity, reframing sarcoidosis as a disorder of disrupted host–commensal homeostasis rather than a classical infectious or autoimmune disease. By integrating microbiological, immunological, and pathological evidence, this review synthesizes the mechanistic basis of EHI and outlines how tolerance failure to *C. acnes* can account for the paradoxical clinical behavior of sarcoidosis. The EHI paradigm further provides a unifying conceptual lens through which related chronic inflammatory disorders—including Crohn’s disease, chronic rhinosinusitis, and atopic dermatitis—may be reinterpreted.

## 1. Introduction

Sarcoidosis is a systemic granulomatous disease characterized by noncaseating epithelioid cell granulomas that develop in multiple organs, most commonly the lungs and lymph nodes. Despite more than a century of investigation, its etiology has remained unresolved [1]. The disorder displays paradoxical features that challenge conventional infectious or autoimmune frameworks: it is noncontagious yet inflammatory, self-limiting in many cases yet chronic or relapsing in others, and histologically uniform despite clinical heterogeneity [2]. These long-standing contradictions indicate that sarcoidosis cannot be fully explained by classical models of infection or autoimmunity, but instead point to a disturbance in the normally stable equilibrium between host immunity and indigenous microbes [3,4].

Accumulating microbiological and immunopathological evidence identifies *Cutibacterium acnes* (formerly *Propionibacterium acnes*), an indigenous skin commensal, as the principal antigenic agent in sarcoidosis. This organism is repeatedly detected in sarcoid lesions by culture [5,6], polymerase chain reaction (PCR) [7,8], in situ hybridization (ISH) [9], and immunohistochemistry (IHC) using species-specific monoclonal antibodies (PAB) [10]. Among these techniques, IHC provides particularly compelling evidence, as PAB-positive signals directly localize *C. acnes* antigens within epithelioid cells and multinucleated giant cells of sarcoid granulomas (Figure 1). Nevertheless, other microorganisms—most notably mycobacteria—have also been proposed as potential contributors to sarcoidosis in subsets of patients, underscoring the heterogeneity of microbial associations reported across studies.

In the immunohistochemical study [10], *C. acnes* antigens were detected by the PAB antibody in 88% of Japanese and 89% of German sarcoid lymph node specimens, whereas all non-sarcoid granulomatous controls were negative. This high sensitivity and complete specificity across ethnically distinct populations strongly indicate that *C. acnes* is not an incidental contaminant but a reproducibly localized antigenic component of sarcoid granulomas, supporting a genuine microbiological association with granuloma formation.

In addition to granuloma-associated bacterial antigens, sarcoid lymph nodes frequently contain Hamazaki–Wesenberg bodies—lysosome-like, spheroidal cytoplasmic inclusions observed within sinus macrophages [11]. These structures are consistently reactive with the *C. acnes*-specific PAB antibody and are considered to represent intracellular reservoirs of *C. acnes*-derived components rather than nonspecific degenerative products [10]. Their frequent presence in sarcoid lymph nodes further supports the concept of latent intracellular persistence of *C. acnes* within the reticuloendothelial system.

To reconcile microbial persistence with noncontagious, hypersensitivity-driven inflammation, we introduce the concept of Endogenous Hypersensitivity Infection (EHI) [3]. This paradigm defines a class of immune disorders that arise from breakdown of tolerance to indigenous microbes persisting intracellularly in a dormant state [12].

EHI addresses a central limitation of classical models: neither infection nor autoimmunity alone can account for the combination of microbial localization, lack of contagion, and antigen-specific hypersensitivity observed in sarcoidosis [13]. It occupies the conceptual domain between these two frameworks, where neither model alone suffices.

EHI posits that endogenous reactivation of a commensal microbe provokes antigen-specific cellular hypersensitivity in genetically and immunologically predisposed hosts, producing granulomatous inflammation without active contagion [14]. Thus, EHI bridges the classical divide between infection and autoimmunity and reframes sarcoidosis as an immune-regulatory failure of host–commensal homeostasis rather than a disease caused by exogenous pathogens.

The historical trajectory leading to the EHI paradigm originates from Japan, where systematic microbiological studies in the late 1970s first implicated *C. acnes* in sarcoidosis. National collaborative research groups isolated *C. acnes* from lymph nodes in a high proportion of sarcoidosis patients and, at lower frequencies, from control samples [5,6]. Because *C. acnes* is a common skin commensal, its recovery from both patient and control specimens initially raised concerns of contamination during biopsy or culture procedures.

However, the markedly higher isolation rate in active sarcoid lesions—together with the absence of any other detectable microorganisms—argued against simple contamination and suggested a biologically meaningful association. Subsequent molecular and immunopathological investigations reinforced this link: quantitative PCR (qPCR) confirmed an increased burden of *C. acnes* genomes in sarcoid tissues [7,8], whereas ISH and IHC localized bacterial nucleic acids and antigens within sarcoid granulomas [9,10]. Together, these convergent lines of evidence established *C. acnes* as the only microorganism consistently detected within sarcoid lesions, thereby providing the empirical foundation for conceptualizing sarcoidosis as an endogenous, commensal-associated process central to the EHI model.

At the immunological level, EHI views sarcoidosis as a disorder of immune regulation rather than microbial invasion [4]. Under normal conditions, host–commensal coexistence is maintained by mechanisms that enforce peripheral tolerance to commensal-derived antigens, including regulatory T-cell (Treg) activity, autophagy-mediated antigen handling, and anti-inflammatory signaling [15]. In sarcoidosis, these checkpoints appear partially compromised: defective autophagy in macrophages [16] and diminished Treg function [17] lead to prolonged antigen presentation and sustained Th1/Th17 polarization [18], an immunological profile consistent with recent analyses of sarcoidosis immunopathogenesis [19].

Once reactivated, dormant *C. acnes* within macrophages serves as a persistent source of endogenous antigens, driving granuloma formation even in the absence of active infection [20]. Such antigen-dependent granuloma maintenance parallels contemporary models describing sarcoidosis as an antigen-driven inflammatory disorder [21].

This immune milieu resembles a controlled hypersensitivity rather than uncontrolled infection—an adaptive but maladjusted attempt to contain reactivated symbionts, consistent with emerging frameworks that interpret sarcoidosis as a disorder of immune homeostasis [22]. Thus, EHI provides a mechanistic framework that unifies microbial persistence, tolerance failure, and exaggerated cellular immunity, positioning sarcoidosis as an immune homeostasis disorder arising from disrupted host–commensal balance.

This review consolidates current evidence supporting the role of *C. acnes* in sarcoidosis and interprets these findings within the EHI framework. By integrating microbiological, immunological, and pathological perspectives, we clarify how latent infection by an indigenous commensal can evoke hypersensitivity-driven granulomatous inflammation through failure of immune tolerance. We further highlight the immunopathological mechanisms that differentiate EHI from conventional models of infection or autoimmunity and demonstrate how this paradigm resolves the clinical and pathological paradoxes of sarcoidosis.

Finally, we extend the EHI framework to related chronic inflammatory disorders involving aberrant responses to commensal microbes and outline future directions for improving diagnosis and developing tolerance-based therapeutic strategies grounded in this conceptual model.

## 2. Evolution of the Concept of Endogenous Infection

The idea that microorganisms could cause disease emerged in the late nineteenth century with the rise of bacteriology [23]. Robert Koch’s formulation of his famous postulates provided a rigorous framework for establishing causal links between pathogens and specific diseases, focusing primarily on exogenous infection—where a foreign organism invades the host and produces pathology [24,25]. This approach proved extraordinarily successful for many acute infectious diseases such as tuberculosis and cholera [26]. However, from the earliest days of bacteriology, several investigators also recognized that some microbes coexisted with humans as part of the normal flora and might, under certain conditions, contribute to disease endogenously without any external invasion [27,28,29,30,31].

In 1886, the Austrian pediatrician Theodor Escherich published *Die Darmbakterien des Säuglings und ihre Beziehungen zur Physiologie der Verdauung* (The Intestinal Bacteria of Infants and Their Relation to the Physiology of Digestion), a seminal monograph describing the bacterial inhabitants of the infant intestine [32,33]. His meticulous microscopy and culture studies revealed that certain bacterial species were constant companions of healthy individuals, forming what we now call the intestinal microbiota [34,35]. Escherich’s work not only led to the naming of *Escherichia coli* but also established the principle that symbiotic microbes are integral to normal human physiology [36]. This idea—radical for its time—laid the foundation for distinguishing benign commensals from disease-causing exogenous pathogens [28].

In the early twentieth century, German-speaking clinicians introduced the terms *endogene Infektion* (endogenous infection) and *Selbstinfektion* (self-infection) to describe infections that arose from microbes already residing within the host. Pankow (1912) discussed such cases in obstetrics and gynecology, emphasizing that microorganisms from the genital tract could become pathogenic under altered physiological conditions [37]. Heidler (1924) similarly elaborated on the mechanisms of self-infection in surgical wounds and postpartum sepsis [38]. These early discussions reflected a growing recognition that disease could arise not from exogenous invasion but from internal shifts in host–microbe relations, a concept remarkably aligned with modern understandings of microbial pathogenicity [28,36].

In the 1910s–1930s, the notion of internal microbial foci gained traction in English-speaking medicine through the “focal infection” theory [39]. Surgeons and physicians such as William Hunter (1921) and Frank Billings (1912) proposed that chronic diseases including arthritis, nephritis, and endocarditis could originate from latent bacterial foci in the tonsils, teeth, or appendix [40,41]. The hypothesis spurred widespread tonsillectomies and dental extractions in the 1920s, but later evidence showed that such interventions rarely improved systemic disease [42]. The focal infection movement consequently fell into disrepute by mid-century [23]. Nevertheless, the idea that internal reservoirs of commensal microbes might drive chronic inflammation anticipated later concepts of microbial persistence, host–microbe imbalance, and the possibility of endogenous reactivation [28,36,43,44].

The discovery of penicillin and the ensuing “antibiotic revolution” of the 1940s exemplified the triumph of Koch’s exogenous paradigm. As summarized by Hutchings et al. (2019), the unprecedented success of antimicrobial therapy in this era entrenched the view that infectious disease equated to invasion by external microbial agents [45]. Endogenous or commensal-derived infections were largely marginalized as clinically unimportant or mere opportunistic complications. The prevailing view for much of the twentieth century was that health depended on eliminating microbes rather than understanding or maintaining host–microbe coexistence [28,36,46,47,48].

Advances in molecular biology and sequencing in the 1990s and 2000s transformed this perspective. The human microbiome project revealed that the human body harbors trillions of microorganisms forming intricate ecological networks [49]. Studies by Chow et al. [50] introduced the term “pathobiont”—a commensal microbe capable of inducing pathology under dysregulated conditions—while Belkaid and Hand demonstrated that immune homeostasis depends on continuous tolerance toward commensals [15,51]. Byrd and Segre [52] revisited Koch’s postulates in light of these discoveries, arguing that microbial context and host susceptibility, rather than mere presence of a pathogen, determine disease outcome [53]. Collectively, these advances reframed endogenous infection as a biologically plausible and clinically relevant process rooted in disrupted host–commensal interactions.

Clinical research also began to revisit diseases long considered “idiopathic.” Work in hepatology, gastroenterology, and hospital medicine showed that chronic liver disease, inflammatory bowel disease (IBD), and hospital-acquired infections could, under altered host conditions, originate from a patient’s own microbiota [54]. Although many of these examples reflect opportunistic or dysbiosis-driven processes rather than true endogenous reactivation, they highlighted a broader principle: resident microbes can contribute to disease when host–commensal interactions become disturbed. In dentistry and periodontology, Rocca (2020) similarly reevaluated the focal infection hypothesis using modern microbial and immunological evidence [43]. These developments paved the way for contemporary models in which specific commensals—under defined immunological and microbial contexts—may drive chronic inflammation, a concept expanded further in Section 4.

As this renewed understanding of endogenous infection gained momentum, *C. acnes* emerged as the most consistently implicated indigenous microbe in sarcoidosis. Findings from culture, qPCR, ISH, and IHC uniformly demonstrated its localization within granulomatous lesions and its ability to provoke Th1-skewed cellular immune responses in affected patients. These convergent data established a reproducible association between *C. acnes* and sarcoid pathology, providing the empirical basis for reframing sarcoidosis as a disorder arising from disrupted host–commensal interactions.

On this basis, Eishi (2013) proposed that sarcoidosis represents an “allergic (hypersensitive) endogenous infection” driven by *C. acnes*, a ubiquitous skin commensal [20]. A decade later, Eishi (2023) expanded this idea into the broader concept of Endogenous Hypersensitivity Infection (EHI), encompassing diseases in which immune tolerance to commensal microbes breaks down, leading to antigen-specific hypersensitivity responses [3]. This framework unites historical insights with contemporary microbiology and immunology, redefining endogenous infection not as microbial overgrowth but as a host-centered failure of immune regulation and containment.

The concept of endogenous infection has thus evolved through three major transformations: (1) its initial recognition in early twentieth-century clinical medicine; (2) its decline and marginalization during the antibiotic era; and (3) its revival and reinterpretation through advances in microbiome and immunology research. The EHI paradigm represents the culmination of this evolution—transforming a once-neglected historical notion into a mechanistic, immunologically grounded framework for understanding host–commensal interactions. This historical trajectory—from the early idea of “self-infection” to its modern immunological reinterpretation—is summarized chronologically in Table 1, which outlines key milestones in the development of the endogenous infection concept leading to the contemporary EHI framework.

## 3. Conceptual Framework of Endogenous Hypersensitivity Infection

Building directly on the historical and experimental foundations summarized in Section 2, the concept of EHI provides a modern immunological interpretation of how commensal microbes can act as conditional pathogens when immune tolerance breaks down. In framing this perspective, this section integrates microbiological, immunological, and clinical insights to articulate a coherent conceptual model of EHI. This model occupies an intermediate position between classical infection and autoimmunity, combining defining features of both paradigms, as summarized in Table 2.

Koch’s postulates, formulated in the late nineteenth century, defined infection as a process initiated by exogenous pathogens invading a susceptible host. While this paradigm revolutionized medicine by identifying specific microbial causes for diseases such as tuberculosis and cholera, its explanatory power becomes increasingly limited when applied to chronic, relapsing, or multifactorial disorders in which consistent isolation of a single pathogen is impossible [56]. Modern microbiome research has demonstrated that health depends on a dynamic equilibrium between the host and vast communities of commensals, and that disruption of this equilibrium—particularly through loss of immune tolerance or dysregulated host responses—can precipitate inflammation [50,57]. The EHI model builds on this insight, reframing infection not as a binary presence or absence of an exogenous pathogen but as a spectrum of host–microbe interactions shaped by immune regulation and microbial persistence.

EHI is defined as a pathogenic condition in which immune responses are aberrantly directed against indigenous microbes that normally persist in a state of immunological tolerance [3,4,20]. Its core features can be conceptualized as follows. First, commensal microbes frequently maintain persistent or latent intracellular residence and may undergo endogenous reactivation in response to specific environmental, metabolic, or immune perturbations [17,18,58]. Second, the disorder reflects a selective breakdown of immune tolerance—most often involving impaired Treg function, altered antigen presentation pathways, or defects in autophagy-mediated microbial processing [59,60,61]. Third, these events give rise to context-dependent chronic immunopathology in which limited intracellular reactivation and sustained antigen release drive antigen-specific hypersensitivity rather than overt microbial expansion [57]. Finally, remission can occur when immune tolerance is re-established and the host regains effective control over commensal antigens [50].

Thus, unlike classical infections that depend on microbial virulence and proliferation, EHI represents a fundamentally host-centered disequilibrium in which dysregulated immune reactivity converts ordinarily harmless commensals into conditional pathogens. In this framework, endogenous reactivation denotes a shift from latent intracellular persistence to a constrained replicative state sufficient to increase antigen availability and trigger hypersensitivity responses [17,18].

Mechanistically, multiple pathways converge to disrupt immune tolerance in EHI. Genetic susceptibility shapes the baseline architecture of antigen presentation and immune regulation. In sarcoidosis, HLA-DRB1*03 and DRB1*15 demonstrate reproducible associations with disease susceptibility and clinical phenotype [62,63], whereas in Crohn’s disease, the autophagy-related variant ATG16L1 (T300A)—rather than ATG5—confers a consistently replicated risk [64,65]. Beyond genetic determinants, immune tolerance is programmed during critical developmental windows in early life, when microbial exposures calibrate long-term regulatory circuits; perturbations during these periods—such as altered skin or mucosal colonization, excessive hygiene, or early antibiotic use—may predispose individuals to later tolerance failure [4,66]. In parallel, autophagy dysfunction in sarcoidosis macrophages compromises intracellular clearance of commensals such as *C. acnes*, fostering microbial persistence and sustained antigen release [16,67,68,69].

Failure of these homeostatic systems permits commensal-derived antigens to persist within macrophages and dendritic cells, driving sustained Th1/Th17 activation and granulomatous inflammation [70,71]. The dynamic sequence of events characterizing EHI can thus be conceptualized as a cyclical process linking microbial persistence, immune reactivation, and tolerance breakdown (Figure 2). Table 3 summarizes this proposed mechanistic progression, using sarcoidosis as a representative example of how latent *C. acnes* reactivation can precipitate antigen-specific, hypersensitivity-driven granulomatous inflammation.

EHI thus bridges the conceptual divide between infection and autoimmunity by introducing a “third category” of disease. Within this continuum, classical infections occupy one pole, defined by microbial invasion and virulence, whereas autoimmune diseases reside at the opposite pole, characterized by self-directed immunity in the absence of microbes. EHI lies between these extremes: immune failure targets non-self yet indigenous antigens derived from commensals. This framework aligns with the contemporary concept of pathobionts—resident microbes that remain harmless under conditions of tolerance but become inflammatory when host regulatory mechanisms fail [50]. It also provides a unifying perspective on chronic inflammatory conditions in which aberrant immune responses to commensal organisms have been increasingly recognized, including sarcoidosis, Crohn’s disease [72], atopic dermatitis [73], and chronic rhinosinusitis [74].

Within the EHI paradigm, clinical heterogeneity reflects the dynamic balance between microbial persistence and host immune tolerance. Spontaneous remission, a hallmark of sarcoidosis, likely corresponds to restoration of tolerance and/or reduction in latent microbial reservoirs, whereas chronic or relapsing disease implies intermittent antigen release from persistent commensals under impaired host regulation. This endogenous activation–tolerance dynamic accounts for fluctuations between remission and relapse in the absence of external reinfection and helps explain why classical antimicrobial therapy alone is often insufficient. Accordingly, therapeutic strategies that restore immune tolerance—such as enhancement of regulatory pathways (e.g., Treg function) or correction of degradative defects (e.g., autophagy dysfunction)—are mechanistically positioned to induce durable remission even without complete microbial eradication. This balance between tolerance restoration and recurrent endogenous activation can be conceptualized as two immunoregulatory phenotypes within the EHI framework—self-remitting and refractory–relapsing forms of sarcoidosis—as summarized in Table 4.

Experimental studies further substantiate this clinical spectrum. In mice and rabbits immunized with heat-killed or recombinant *C. acnes* antigens, sarcoid-like pulmonary granulomas develop even in healthy hosts, confirming that hypersensitivity to an indigenous microbe can drive granulomatous inflammation [20,59,75]. These lesions are self-limiting and, after resolution, become refractory to antigenic re-challenge—mirroring the spontaneous remission and restoration of tolerance observed in many sarcoidosis patients. Conversely, when tolerance fails to recover, persistent or relapsing inflammation ensues, analogous to chronic, therapy-resistant sarcoidosis. Antibiotic pretreatment prevents disease induction in these models, indicating that latent endogenous bacteria serve as the antigenic source rather than exogenous infection [20,75]. Importantly, although *C. acnes* sensitization in these models is performed using Freund’s complete adjuvant, control animals immunized with saline emulsified in the same adjuvant do not develop pulmonary granulomas. This demonstrates that mycobacterial components within the adjuvant are not sufficient to induce granulomatous inflammation, and that granuloma formation depends on antigen-specific immune responses to *C. acnes*. Notably, conventionally housed rabbits with greater baseline pulmonary *C. acnes* load exhibit more extensive and confluent granulomas than SPF mice, suggesting that microbial burden modulates the hypersensitivity threshold (Figure 3) [20]. These findings collectively validate EHI as a tolerance-failure spectrum ranging from self-remitting to refractory–relapsing forms, paralleling the clinical diversity of sarcoidosis.

Taken together, EHI represents a distinct disease paradigm in which breakdown of immune tolerance to latent indigenous microbes culminates in chronic immune-mediated inflammation. Upon endogenous reactivation, intracellular *C. acnes* undergoes limited replicative activity that increases antigen release, thereby eliciting a Th1-dominant hypersensitivity response in genetically and immunologically susceptible hosts. Sarcoidosis serves as the most illustrative prototype of this framework, in which latent intracellular *C. acnes* functions as a persistent antigen source—rather than a fully replicating pathogen—to drive systemic granulomatous inflammation through dysregulated host–microbe interactions. By clarifying these core mechanisms, this prototype provides a conceptual foundation for extending the EHI framework to other chronic inflammatory disorders driven by aberrant immune responses to commensal antigens. Importantly, this framework also offers a coherent explanation for the marked inter-individual and population-level heterogeneity that characterizes sarcoidosis.

In this context, ethnic and organ-specific differences emerge as modifiers of disease expression rather than as evidence of distinct pathogenic mechanisms. Sarcoidosis exhibits marked variability in organ involvement, clinical course, and prognosis among individuals and across populations, a feature that has long complicated etiologic interpretation. From an EHI perspective, such heterogeneity is not unexpected but rather reflects differences in host genetic background, immune recognition pathways, and environmental context that shape tolerance or hypersensitivity to endogenous microbial antigens. Variations in HLA haplotypes, innate immune receptors, and antigen-processing mechanisms may influence how commensal-derived antigens such as those from *C. acnes* are presented and perceived by the host immune system. Likewise, tissue-specific factors—such as local macrophage populations, antigen load, and microenvironmental immune regulation—may determine whether loss of tolerance manifests predominantly in the lung, lymph nodes, skin, eye, or other organs. Thus, EHI provides a conceptual framework that unifies the microbial association with the pronounced inter-individual and ethnic diversity characteristic of sarcoidosis, without invoking fundamentally different disease mechanisms. Within this context, the focus on *C. acnes* does not preclude the involvement of other microbial candidates, including mycobacteria, but reflects the current strength of lesion-based and immunopathological evidence supporting *C. acnes* as the most consistent prototype of sarcoidosis within the EHI framework.

## 4. Broader Relevance of the EHI Concept

While sarcoidosis provides the clearest and most extensively characterized example of an EHI, the core elements of this framework—microbial persistence, breakdown of immune tolerance, and hypersensitivity-driven inflammation—are increasingly recognized across a wide range of chronic inflammatory disorders. This perspective is consistent with broader conceptual advances in microbiota–host interactions [76] and with evidence implicating commensal microbes in chronic immune-mediated inflammation, particularly in inflammatory bowel disease (IBD) [77]. Viewed through the EHI lens, these conditions can be understood as variations on a shared pathogenic theme rather than as isolated disease entities.

### 4.1. Inflammatory Bowel Disease (IBD)

IBD represents a prototypical example of chronic inflammation driven by dysregulated host–microbiota interactions [60]. Exaggerated T-cell responses to intestinal commensals, together with genetic susceptibility variants affecting microbial sensing and autophagy pathways (e.g., NOD2, ATG16L1, IRGM), underscore the importance of impaired intracellular bacterial handling in disease pathogenesis [78,79]. Although sarcoidosis is associated with NOD1 rather than NOD2 variants [80], both conditions converge on disrupted intracellular pattern-recognition and tolerance mechanisms. In this sense, Crohn’s disease may be regarded as an intestinal counterpart to sarcoidosis, with site-specific manifestations reflecting failure of tolerance to distinct indigenous microbes within the EHI framework.

### 4.2. Chronic Rhinosinusitis (CRS)

Chronic rhinosinusitis with nasal polyps illustrates how localized failure of mucosal immune tolerance can result in persistent inflammation. Commensal organisms such as *Staphylococcus aureus* are frequently detected in affected tissues, where superantigenic and biofilm-associated factors amplify immune activation [81,82,83]. The persistence of disease despite antimicrobial therapy, together with skewed Th2/Th17 responses and impaired regulatory control [84,85,86], supports the concept that pathology arises primarily from hypersensitivity to resident microbes rather than from ongoing infection, in close alignment with EHI principles.

### 4.3. Atopic Dermatitis (AD)

Atopic dermatitis exemplifies a cutaneous manifestation of commensal-driven hypersensitivity. Disruption of the skin barrier facilitates colonization by *Staphylococcus aureus* and abnormal immune exposure to microbial antigens [87,88,89]. The resulting Th2- and Th17-biased inflammation, coupled with impaired regulatory T-cell function [90,91], highlights a breakdown of host–microbiome equilibrium. The chronic, relapsing course of AD and the limited durability of antimicrobial interventions further indicate that disease activity is sustained by immune hypersensitivity to colonizing commensals rather than by active infection [92].

### 4.4. SAPHO Syndrome and Localized Commensal Infections

The EHI framework can also be extended to selected musculoskeletal disorders. SAPHO syndrome has been repeatedly associated with *C. acnes*, which has been detected within osteoarticular lesions and is thought to act as a persistent endogenous inflammatory trigger in susceptible hosts [93,94]. In contrast, other *C. acnes*-associated conditions—such as acne vulgaris [95], implant-associated osteitis [96], and chronic prostatitis [97,98,99,100]—are more appropriately interpreted as site-restricted manifestations of commensal persistence or localized immune activation. These disorders typically lack the systemic tolerance failure and multiorgan involvement that define EHI, underscoring the distinction between localized commensal infections and systemic hypersensitivity phenotypes.

### 4.5. Conceptual Integration and Implications

Taken together, the disorders discussed above share a recurring pathogenic motif: persistence of commensal microbes, failure of immune tolerance, and exaggerated immune responses leading to chronic inflammation. Although clinical manifestations differ by tissue context, the underlying host–commensal interaction follows analogous principles. Framing this spectrum of diseases within the EHI paradigm helps dissolve traditional boundaries between infectious and autoimmune categories and instead emphasizes immune tolerance to indigenous microbes as a central determinant of disease. Representative examples of EHI-like mechanisms across chronic inflammatory disorders are summarized in Table 5.

## 5. Clinical and Translational Implications

The EHI framework, initially developed through studies of *C. acnes*-associated sarcoidosis [3,4], provides a unified conceptual basis for interpreting the clinical, pathological, and immunological features of this disease. By transcending the traditional dichotomy between infection and autoimmunity, EHI emphasizes disease as a consequence of regulatory imbalance at the host–commensal interface rather than uncontrolled microbial invasion [55,101]. This relational view closely aligns with the host-centered paradigm proposed by Casadevall and Pirofski, which defines pathogenicity as an emergent outcome of host–microbe interactions [102,103], and with the concept of disease tolerance that prioritizes limitation of immunopathology over pathogen eradication [104].

Within this framework, sarcoidosis emerges not as an idiopathic granulomatous disorder but as a prototype of endogenous infection, in which context-dependent hypersensitivity to an indigenous microbe drives chronic inflammation [103]. These conceptual advances have direct clinical and translational relevance, reframing both diagnosis and treatment around the identification of microbial persistence and restoration of immune tolerance rather than reliance on exclusion-based criteria or nonspecific immunosuppression.

### 5.1. Diagnostic Applications

Current diagnostic criteria for sarcoidosis rely primarily on morphological findings and exclusion of alternative causes, approaches that do not elucidate disease etiology [105]. The EHI framework instead highlights lesion-based strategies that directly detect microbial antigens within granulomas [106,107], together with immunologic profiling that reflects tolerance failure toward *C. acnes* [108,109,110]. Immunohistochemical detection of *C. acnes* within sarcoid granulomas using the PAB monoclonal antibody, which recognizes species-specific membrane-anchored lipoteichoic acid of *C. acnes*, represents the most specific tissue-based indicator of microbial involvement and complements molecular approaches such as quantitative PCR.

Beyond tissue detection, EHI emphasizes antigen-specific immune responses as diagnostically informative. Heightened *C. acnes*-specific Th1 responses [111,112,113], the presence of immune complexes containing *C. acnes* antigens in lymph nodes and circulation [114,115], and quantitative or functional impairments of FOXP3^+^ regulatory T cells [116,117] collectively indicate a breakdown of immune tolerance. Systematic assessment of these parameters may yield mechanism-based biomarkers that complement established clinical indices and enable more precise disease stratification [118,119].

### 5.2. Therapeutic Implications

Conventional treatment of sarcoidosis relies largely on corticosteroids and nonspecific immunosuppressants, which alleviate inflammation but do not reliably restore durable immune stability [1,120,121]. In contrast, the EHI model reframes therapy as restoration of immune tolerance toward a persistent commensal rather than eradication of a classical pathogen. Within this context, *C. acnes* functions as a pathogenic trigger only when regulatory mechanisms fail, underscoring the central role of Tregs and autophagy-dependent intracellular control pathways in maintaining host–microbe homeostasis [67,80,122,123,124,125,126].

Therapeutic strategies that enhance these regulatory circuits therefore represent rational, mechanism-based interventions. Agents that promote autophagy and regulatory immune function—such as rapamycin, metformin, and low-dose interleukin-2—have demonstrated the capacity to attenuate proinflammatory signaling while expanding or stabilizing Treg populations [127,128,129,130,131,132,133]. Antimicrobial therapies, including those evaluated in the CLEAR trial [134] and the use of minocycline in cutaneous sarcoidosis [135], may reduce antigenic burden or microbial activity rather than eradicate replicating organisms, thereby complementing tolerance-restoring approaches.

Sustained remission is likely to require simultaneous reduction in microbial persistence and reinforcement of host regulatory mechanisms. Emerging strategies, such as antigen-specific immunomodulation, adoptive transfer of *C. acnes*-specific Tregs, and microbiota-targeted interventions [136,137], further illustrate the translational potential of EHI-guided therapy. The key diagnostic and therapeutic implications of this tolerance-centered framework are summarized in Table 6, highlighting the shift from empirical immunosuppression to mechanistically guided, host-centered intervention.

## 6. Conclusion and Future Directions

The EHI paradigm represents a conceptual advance that bridges the long-standing divide between infection and autoimmunity. For more than a century, medical science has been shaped by Koch’s exogenous infection model, which attributes disease to invasion by external pathogens, while immune-mediated disorders were viewed as arising from aberrant self-reactivity. Yet a substantial group of chronic inflammatory conditions—including sarcoidosis, IBD, and CRS—cannot be adequately explained by either constructs alone. These disorders share a unifying pathophysiological principle: the breakdown of immune tolerance toward commensal or latent microbes residing within the host.

By defining EHI as a distinct pathogenic category, we highlight that disease arises not from microbial virulence per se, but from the host’s failure to maintain immunoregulatory control over its indigenous microbiota. This host-centered model reframes the relationship between infection and immunity as a dynamic continuum governed by tolerance rather than a binary confrontation between “self” and “non-self.” Within this conceptual space, EHI offers a unified framework that links microbial persistence, immune dysregulation, and chronic inflammation across organ systems. More broadly, it is consonant with contemporary perspectives on host–microbe coevolution and the recognition of the microbiome as an integral component of immune homeostasis.

The introduction of EHI as a formal disease construct carries far-reaching implications for biomedical research. It encourages the re-examination of long-standing “idiopathic” or “autoimmune” disorders through a new mechanistic lens and guides the development of diagnostic and therapeutic strategies aimed at restoring immune tolerance rather than merely suppressing inflammation. In this sense, the EHI paradigm may provide a conceptual foundation for a post-Kochian era of medicine—one that seeks to re-establish equilibrium with commensal life rather than pursue its eradication.

Sarcoidosis has long stood as one of the most enigmatic challenges in clinical medicine, defying classical infection theory and eluding classification as a purely autoimmune disorder. The concept of EHI provides a unifying paradigm that helps resolve many of these longstanding paradoxes. By reframing sarcoidosis as a prototype of EHI, we underscore that its pathogenesis arises from a selective breakdown of immune tolerance to an indigenous commensal—most prominently *C. acnes*—in genetically and immunologically susceptible hosts.

The EHI framework helps resolve several long-standing enigmas of sarcoidosis, including the coexistence of spontaneous remission and chronic refractory progression, as well as the striking histological uniformity of granulomas despite pronounced clinical heterogeneity. It further accommodates the presence of latent microbial reservoirs, such as Hamazaki–Wesenberg bodies, within a unified pathogenic model. By integrating Treg dysfunction, impaired autophagy, genetic susceptibility, and potential early-life failures of microbial tolerance, EHI provides a coherent and biologically plausible framework for understanding disease persistence and variability.

Beyond sarcoidosis, the EHI paradigm broadens our capacity to interpret a range of chronic inflammatory disorders—including IBD, CRS, and AD—in which immune tolerance to commensal microbes is compromised. In this way, it helps bridge the long-standing conceptual divide between classical infection and autoimmunity, delineating a third pathogenic category that more accurately reflects clinical and biological reality.

Looking ahead, recognition of EHI as a distinct pathogenic framework carries important implications for both research and clinical practice. It calls for the development of antigen-specific biomarkers and tolerance-restoring therapies, thereby shifting clinical management away from nonspecific immunosuppression toward mechanism-based precision immunomodulation. This perspective also motivates new clinical trial designs grounded in mechanistic endpoints, such as the restoration of immune tolerance or the reduction in latent microbial reservoirs. Ultimately, formalizing EHI as a disease category has the potential to reshape our understanding of a broad spectrum of chronic inflammatory disorders, advancing both basic science and translational medicine. Although this immunoregulatory model remains hypothetical, it provides a coherent conceptual structure that links immune dysregulation to the clinical heterogeneity characteristic of sarcoidosis.

In conclusion, the EHI paradigm represents a fundamental shift in our understanding of infection, immunity, and host–microbe relationships. By positioning sarcoidosis as its prototype, this framework not only addresses a century-old etiological question but also provides a conceptual foundation for rethinking chronic inflammatory diseases across medicine. Ongoing advances in genomics, single-cell profiling, and immunophenotyping are expected to refine patient stratification within the EHI spectrum, enabling the identification of subgroups defined by distinct microbial or immunologic signatures. As with other major shifts in disease pathogenesis, the EHI framework is expected to mature through incremental advances and the accumulation of convergent evidence across experimental, clinical, and translational studies. Future clinical trials guided by these mechanistic insights—using endpoints such as the re-establishment of immune tolerance or the reduction in latent microbial antigens—will be essential for validating the EHI model and for advancing precision immunotherapy for sarcoidosis and related disorders.

## Figures and Tables

**Figure 1 microorganisms-14-00147-f001:**
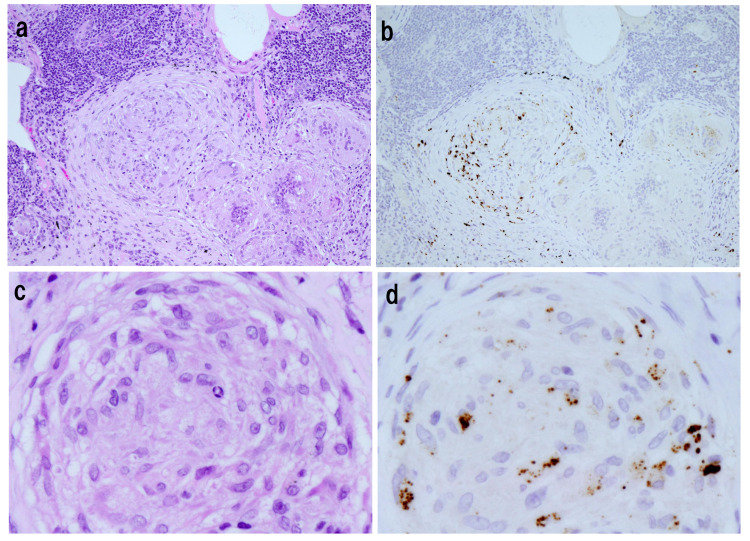
Immunohistochemical detection of *Cutibacterium acnes* antigens in sarcoid granulomas using the *C. acnes*-specific monoclonal PAB antibody. Positive brown granular signals are visible within epithelioid cells and multinucleated giant cells, confirming the presence of bacterial components within sarcoid granulomas. Panels (**a**,**b**) show splenic sarcoidosis (original magnification ×100), and panels (**c**,**d**) show pulmonary sarcoidosis (original magnification ×400). All images are original and previously published [3].

**Figure 2 microorganisms-14-00147-f002:**
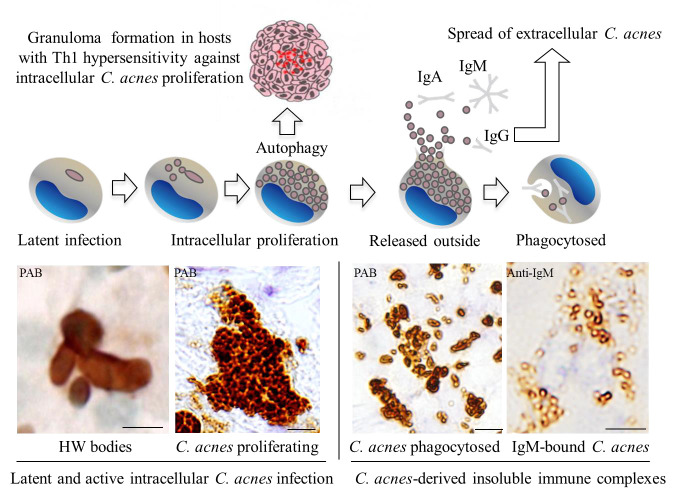
Proposed pathogenic cycle of *Cutibacterium acnes* in sarcoidosis. Latent intracellular *C. acnes* may undergo endogenous reactivation under immunologic or metabolic stress, leading to limited intracellular proliferation and subsequent granuloma formation. Proliferating bacteria can then be released into the extracellular space, where they form IgM- or IgA-bound immune complexes. These immune complexes are phagocytosed by macrophages, establishing new intracellular latent infection. This cyclical process integrates microbial persistence, antigen-specific hypersensitivity, and failure of immune tolerance—the core mechanisms defining Endogenous Hypersensitivity Infection (EHI). All components of the schematic are original and were previously published [3,4].

**Figure 3 microorganisms-14-00147-f003:**
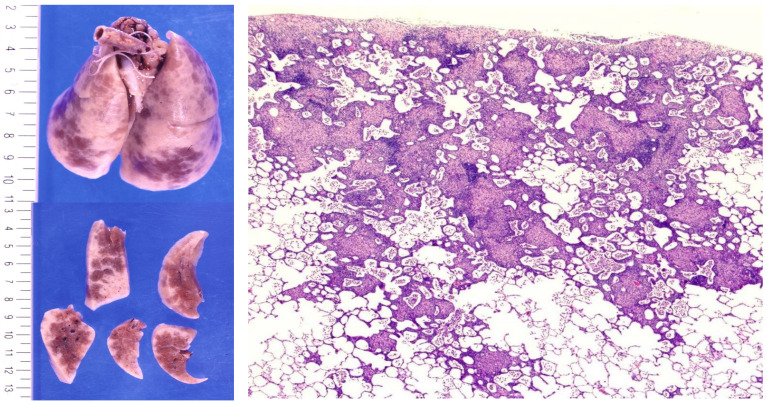
Experimental pulmonary granulomas induced by *Cutibacterium acnes* sensitization in conventionally housed rabbits. Gross images of rabbit lungs (**left**) show multiple whitish granulomatous lesions predominantly distributed in the subpleural and interlobular regions. Corresponding low-power histological sections ((**right**), H&E, ×4) show confluent non-caseating epithelioid granulomas with surrounding lymphocytic infiltration, widely distributed throughout the lung parenchyma following *C. acnes* sensitization with adjuvant. The extent and confluence of granulomas are greater than those typically observed in SPF mice, consistent with higher baseline latent *C. acnes* burden in conventionally housed rabbits. All images are original and previously published [20].

**Table 1 microorganisms-14-00147-t001:** Historical evolution of the concept of endogenous infection.

Era/Period	Key Figures/Publications	Conceptual Focus	Relevance to EHI Framework
Late 19th century (1880s)	Theodor Escherich (1886) [32] *Die Darmbakterien des Säuglings…*	Discovery of intestinal commensals; distinction between symbionts and pathogens	Established the idea that indigenous microbes play physiological roles.
Early 20th century (1910s–1920s)	Pankow (1912) [37]; Heidler (1924) [38]	*Endogene Infektion*/*Selbstinfektion*—disease from resident microbes	Introduced endogenous infection arising from internal microbial imbalance.
1910s–1930s	Hunter (1921) [40]; Billings (1912) [41]	Focal Infection Theory	Linked local microbial persistence to systemic chronic inflammation.
1940s–1960s (Antibiotic era)	Penicillin era; dominance of Koch’s paradigm	Exogenous infection emphasized; endogenous infection marginalized	Shifted medicine toward eradication of microbes, neglecting tolerance.
1990s–2000s (Microbiome era)	Belkaid & Hand [15]; Honda & Littman [55]	Commensals as immunoregulatory partners; pathobiont concept	Reintroduced tolerance and host–microbe balance as essential for health.
2010s–2020s (EHI concept)	Eishi (2013, 2023) [3,20]	EHI—immune tolerance failure to latent commensals	Integrates classical and modern insights into a host-centered framework.

Chronological summary of the intellectual evolution from early concepts of “self-infection” to the modern Endogenous Hypersensitivity Infection (EHI) paradigm.

**Table 2 microorganisms-14-00147-t002:** Conceptual position of Endogenous Hypersensitivity Infection (EHI).

Feature	Classical Infection	Endogenous Hypersensitivity Infection (EHI)	Autoimmunity
Etiologic agent	Exogenous pathogen	Endogenous commensal microbe	None (self-antigen)
Immune target	Non-self	Semi-self (commensal antigens)	Self
Transmission	Contagious	Non-contagious	Non-contagious
Dominant response	Pathogen elimination	Hypersensitivity to commensals	Self-directed immunity
Therapeutic focus	Antimicrobial therapy	Immune regulation ± antimicrobial	Immunosuppression

Comparison between classical infection, autoimmunity, and Endogenous Hypersensitivity Infection (EHI) as three distinct pathogenic frameworks.

**Table 3 microorganisms-14-00147-t003:** Mechanistic process of Endogenous Hypersensitivity Infection (EHI) exemplified by sarcoidosis.

Component	Description	Consequence
Latent reservoirs	Intracellular persistence of *C. acnes* within macrophages	Subclinical latency
Endogenous reactivation	Reactivation under immune or metabolic stress	Limited intracellular proliferation
Antigen presentation	APCs present *C. acnes* antigens	Th1/Th17 activation
Effector axis	Dominant Th1/Th17 polarization	Granuloma formation
Tolerance axis	Treg dysfunction + impaired autophagy	Failure to terminate responses
Outcome	Formation of new latent foci	Relapsing or persistent disease

Cyclical progression linking microbial persistence, hypersensitivity activation, and tolerance failure in Endogenous Hypersensitivity Infection (EHI). Abbreviations: APC, antigen-presenting cell; Th, T helper cell; Treg, regulatory T cell.

**Table 4 microorganisms-14-00147-t004:** Immunoregulatory model distinguishing self-remitting and refractory–relapsing sarcoidosis.

Feature	Self-Remitting Type	Refractory–Relapsing Type
Latent bacterial load	Small, contained	Large, recurrently reactivated
Treg competence	Preserved	Impaired
Effector activity	Transient	Persistent
Autophagy	Functional	Defective
Granuloma resolution	Regression	Fibrosis/persistence
Clinical course	Spontaneous remission	Chronic relapse

Reflecting the dynamic balance between microbial persistence and immune tolerance capacity. Abbreviations: Treg, regulatory T cell.

**Table 5 microorganisms-14-00147-t005:** EHI-related immunological and host susceptibility-associated features across chronic inflammatory diseases.

Disease	Commensal Driver	Tolerance Failure	Pattern of Inflammation
Sarcoidosis	*Cutibacterium acnes*	Treg dysfunction; impaired autophagy	Systemic noncaseating granulomas
Crohn’s disease	*Escherichia coli* and other gut pathobionts	NOD2/ATG16L1/IRGM defects	Chronic intestinal granulomatous inflammation
Chronic rhinosinusitis (CRS)	*Staphylococcus aureus*	Failure to control superantigenic stimulation; reduced Treg activity	Persistent nasal polyps; chronic mucosal inflammation
Atopic dermatitis (AD)	*Staphylococcus aureus*	Skin barrier breakdown + Treg dysregulation	Eczematous flares; relapsing inflammation
SAPHO syndrome	*Cutibacterium acnes*	Osteoarticular microbial persistence	Chronic osteitis and hyperostosis

Comparison of tolerance failure across commensal-driven inflammatory diseases within the Endogenous Hypersensitivity Infection (EHI) framework. Abbreviations: Treg, regulatory T cell; NOD2, nucleotide-binding oligomerization domain–containing protein 2; ATG16L1, autophagy-related protein 16-like 1; IRGM, immunity-related GTPase family M protein.

**Table 6 microorganisms-14-00147-t006:** Potential therapeutic strategies for restoring immune tolerance in sarcoidosis within the EHI framework.

Therapeutic Target	Representative Approaches	Proposed Mechanism
Regulatory T cell (Treg) activation	Low-dose IL-2 therapy Adoptive transfer of antigen-specific Tregs IL-10–supportive cytokine strategies	Expands and stabilizes Tregs; suppresses excessive Th1/Th17 activity; promotes resolution of granulomas.
Autophagy enhancement	Rapamycin (mTOR inhibition) Metformin (AMPK activation) Nutritional/metabolic modulation	Restores intracellular degradation of *C. acnes*, reduces antigen release, re-establishes immune homeostasis.
Host–microbe equilibrium modulation	Microbiota-targeted therapy Therapeutic vaccination with attenuated or recombinant *C. acnes* antigens	Rebalances immune recognition of commensals; prevents hypersensitivity activation.
Targeted antimicrobial therapy	Combination antibiotics (e.g., doxycycline, macrolides) CLEAR trial evidence	Reduces latent *C. acnes* burden; macrolides also provide immunomodulation.
Combined immunoregulatory–antimicrobial therapy	Antibiotics + Treg/autophagy enhancers (e.g., macrolide + rapamycin) Sequential microbial clearance + tolerance restoration	Addresses both microbial persistence and tolerance failure; supports durable remission.
Biomarker-guided precision therapy (future direction)	Monitoring *C. acnes* antigen load Th1/Treg ratio Autophagy gene-expression signatures	Enables individualized therapy according to microbial persistence and tolerance status.

EHI-guided therapy focuses not on microbial eradication but on re-establishing immune tolerance and restoring host–microbe equilibrium. Abbreviations: EHI, Endogenous Hypersensitivity Infection; IL-2, interleukin-2; IL-10, interleukin-10; Treg, regulatory T cell; Th, T helper cell; mTOR, mechanistic target of rapamycin; AMPK, AMP-activated protein kinase.

## Data Availability

No new data were created in this study. Data sharing is not applicable to this article.

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
