# Peer review of "Endogenous Hypersensitivity Infection: A Unifying Framework for Cutibacterium acnes-Associated Sarcoidosis"

_microorganisms, 2026, doi:10.3390/microorganisms14010147_

Round 1

Reviewer 1 Report

Comments and Suggestions for Authors

Thank you for interesting reading!

The review is well-written, you argue for the hypothesis in a way so the reader easily understands and draw historical parallells to other concepts, as well as parallells to other diseases. The review seemed long as a first glance but you bring up relevant issues and it was definitely not boring to read, did not seem long at all when I had started reading. I missed one thing, maybe it could be an idea to discuss  more about different organ manifestations and why they differ between patients of different ethnicities. The concept with EHI goes well with these ethnic differences (different HLA-types, environmental factors etc) but I do not think that is clear from the manuscript. Also, the review is focused on C. acnes but there are other microorganisms that have been suggested ti play a role in sarcoidosis and that is not brought up at all. I suggest to add aline about that c acnes is not the only microorganism suggested to play a role in sarcoidosis.

Author Response

Response to Reviewers

I sincerely thank the Editor and the Reviewers for their careful evaluation of my manuscript and for their constructive and insightful comments. I have revised the manuscript accordingly. Main changes are marked in the revised version. Detailed responses to each comment are provided below.

Reviewer 1

Comment 1

The review is well written, but discussion of organ manifestations and ethnic differences could be expanded. The EHI concept fits well with ethnic differences, but this is not clearly discussed.

Response:

I thank the reviewer for this valuable suggestion. I agree that organ tropism and ethnic variability are important features of sarcoidosis that can be coherently interpreted within the EHI framework. I have therefore added a new paragraph to the Discussion to explicitly address how host genetic background (including HLA diversity), immune regulation, and environmental context may shape organ-specific manifestations and ethnic heterogeneity within a unified EHI mechanism.

Manuscript change:

A new paragraph (lines 434-452) has been added to the Discussion section (Section 3), immediately before Section 4, “Broader Relevance of the EHI Concept”, discussing ethnic differences, organ tropism, and host genetic factors as modifiers of EHI expression in sarcoidosis.

Comment 2

The review focuses on C. acnes, but other microorganisms have also been suggested to play a role in sarcoidosis. This should be acknowledged.

Response:

I appreciate this important point. While my review emphasizes C. acnes because of the strength and consistency of lesion-based and immunopathological evidence, I agree that other microbial candidates have been proposed. I have added clarifying statements in both the Introduction and the Discussion to explicitly acknowledge this broader microbial context while explaining the rationale for focusing on C. acnes as the most robust prototype within the EHI framework.

Manuscript change:

A clarifying sentence has been added to the Introduction (lines 51-54) and reiterated briefly in the Discussion (lines 449-452) to acknowledge other proposed microbial candidates.

Reviewer 2

Comment 1

The article is thorough but long and somewhat repetitive, particularly with repeated citation of reference [15].

Response:

I appreciate this careful reading and agree that redundancy should be minimized. I have systematically reviewed the manuscript and reduced repeated citation of reference [15], retaining it only at two conceptually essential locations where it provides foundational context. Redundant uses supporting the same general point were removed to improve concision without altering the reference list.

Manuscript change:

Redundant citations of reference [15] were removed throughout the manuscript, consolidating its use to two key conceptual sections while preserving the original reference list.

Comment 2

Sections 4 and 5 overlap conceptually, with repetition of similar points. Much of the discussion could be shortened, as Table 6 already summarizes key concepts.

Response:

I agree with this assessment. To address redundancy, Sections 4 (“Broader Relevance of the EHI Concept”) and 5 (“Clinical and Translational Implications”) were carefully streamlined. Repetitive explanatory passages were condensed, while all cited references and the core conceptual framework were retained. This restructuring improves clarity and substantially reduces manuscript length.

Manuscript change:

Sections 4 and 5 were revised to eliminate redundant discussion while retaining all references and key concepts, resulting in an approximate 40% reduction in word count across these sections.

Comment 3

In addition to C. acnes, the possible role of non-tuberculous mycobacteria (NTM) in sarcoidosis should be discussed. If animals are challenged with C. acnes and NTM, does this enhance granuloma formation?

Response:

I appreciate this insightful question. In experimental sarcoidosis models, C. acnes sensitization is typically performed using Freund’s complete adjuvant, which contains mycobacterial components and provides a strong Th1-skewing context. Importantly, however, control animals immunized with saline emulsified in the same adjuvant do not develop pulmonary granulomas, indicating that mycobacterial components alone are insufficient to induce granulomatous inflammation. These findings support the conclusion that granuloma formation is driven by C. acnes–specific immune responses rather than nonspecific adjuvant or mycobacterial stimulation. While direct co-challenge experiments with defined NTM species have not yet been systematically performed, I now acknowledge NTM as a potential additional microbial contributor and identify this as an important topic for future investigation in experimental models.

Manuscript change:

A clarifying statement has been added to the Experimental Studies section (lines 410-415) noting that saline plus Freund’s complete adjuvant does not induce pulmonary granulomas, thereby demonstrating that granuloma formation depends on C. acnes–specific immune sensitization rather than adjuvant or mycobacterial components alone. A brief acknowledgment of NTM as a possible additional contributor has also been included in the Introduction (lines 51-54) and the Discussion (lines 449-452).

Comment 4

Changing disease paradigms takes time, and this should be acknowledged.

Response:

I appreciate this insightful perspective and fully agree. I have added a brief reflective statement to acknowledge that shifts in disease pathogenesis paradigms are typically gradual and depend on the accumulation of convergent evidence.

Manuscript change:

A single reflective sentence (lines 657-659) has been added to the concluding paragraph of the manuscript. Specifically, the following sentence was inserted at the end of the Conclusion section: “As with other major shifts in disease pathogenesis, the EHI framework is expected to mature through incremental advances and the accumulation of convergent evidence across experimental, clinical, and translational studies.”

Minor comments

Response:

I thank the reviewer for noting these points. All formatting and reference issues have been corrected.

Manuscript change:

Bracketed references were reformatted as ranges where appropriate (e.g., [5–6], [7–8], [27–31]), and the missing journal title for reference [65] has been added.

Closing statement

I again thank the Editor and Reviewers for their constructive feedback, which has substantially improved the clarity, balance, and rigor of this manuscript. I believe that the revised version addresses all comments and is now more concise, coherent, and robust, and I hope it will be suitable for publication in Microorganisms.

Reviewer 2 Report

Comments and Suggestions for Authors

see attached critique

Author Response

(The authors gave the same response as above.)
